# Assessment of Drying Kinetics, Textural and Aroma Attributes of *Mentha haplocalyx* Leaves during the Hot Air Thin-Layer Drying Process

**DOI:** 10.3390/foods11060784

**Published:** 2022-03-08

**Authors:** Hui-Ling Guo, Ying Chen, Wei Xu, Meng-Tian Xu, Yong Sun, Xue-Cheng Wang, Xiao-Ya Wang, Jing Luo, Hua Zhang, Yao-Kun Xiong

**Affiliations:** 1Department of Pharmaceutics, College of Pharmacy, Jiangxi University of Traditional Chinese Medicine, Nanchang 330004, China; ghl6262@126.com (H.-L.G.); 202081800029@jxutcm.edu.cn (Y.C.); daxiong5927@163.com (W.X.); 201981702012@jxutcm.edu.cn (M.-T.X.); 20142036@jxutcm.edu.cn (X.-C.W.); 20091026@jxutcm.edu.cn (J.L.); 2State Key Laboratory of Food Science and Technology, Nanchang University, Nanchang 330047, China; yongsun@ncu.edu.cn; 3Department of Food Nutrition and Safety, College of Pharmacy, Jiangxi University of Chinese Medicine, Nanchang 330004, China; 20211028@jxutcm.edu.cn

**Keywords:** *Mentha haplocalyx*, hot air thin layer drying, drying kinetics, sensory quality, bioactive attribute

## Abstract

Since *Mentha haplocalyx* leaves are rich in bioactive constitutes, particularly volatile compounds, there are higher demands for high-quality dried medicinal and aromatic peppermint products. This study aimed to assess the drying kinetics of hot air thin layer drying *Mentha haplocalyx* leaves and exploring the effects of hot air-drying temperatures on the textural properties and sensory quality. According to our results, the Midilli model is the best model representing the hot air-drying process. The effective moisture diffusivity (Deff) and activation energy (Ea) of the hot air-drying process were determined as 7.51 × 10^−9^–3.03 × 10^−8^ m^2^/s and 57.98 KJ/moL, respectively. The changes of textural and aromatic profiles of dried *Mentha haplocalyx* leaves were subsequently evaluated by the SEM, GC–MS and E-nose technology. Changes in leaf cellular membrane structures were observed in this study, indicating that the loss of moisture content induced the shrinkage of leaf cells during the hot air-drying process. Moreover, the altered profile of volatile compounds was identified at the different drying temperatures. As a result of the GC-MS analysis, increasing the content of D-carvone from 61.89%, 69.25% and 78.2% resulted in drying temperatures of 35 °C, 45 °C and 55 °C, respectively; while a decreasing trend of other volatile compounds, including D-Limonene, cineole and l-caryophyllene was detected as drying temperature elevated. Finally, the aromatic profile was evaluated by E-nose, and results of the flavor radar fingerprint and PCA showed that aromatic profiles were significantly altered by the drying process. The overall results elucidated that the hot air thin layer drying at 35 °C efficiently improved the final quality of dried *Mentha haplocalyx* leaves by maintaining flavor properties.

## 1. Introduction

Both fresh and dried peppermint (*Mentha piperita* and *Mentha haplocalyx*) leaves are widely used as food and therapeutic ingredients for flavor and fragrance additives, tea products and pharmaceuticals [1,2]. In addition, the peppermint essential oils derived from peppermint leaves are widely produced in the world for the medical, nutraceutical and food applications. As being an aromatic perennial herb, peppermint leaves contain various bioactive components such as phenolics, tannins, resins, pectin, bitter compounds, but more importantly volatile components [3]. As is well known, *Mentha piperita* and *Mentha haplocalyx* are both rich in menthol and menthone, and those are the main bioactive components of peppermint essential oil and are used for culinary and medicinal products [4]. Due to it being rich in these bioactive constitutes, the intake of peppermint-based products such as herbal tea, essential oils and extracts are beneficial for overall health based on its antimicrobial, anti-oxidative, anti-inflammation and immune modulatory attributes [5]. Accordingly, there is a growing global consumer demand for substantial and high-quality peppermint products. 

Being important food flavor ingredients, peppermint leaves can be used not only directly for fresh food, but also as raw materials for preparing their volatile extractions, which are further applied in various beverage and candy production. The feature aroma as well as the health-promoting properties of peppermint leaves have drawn attention in the food industry for decades. *Mentha haplocalyx,* widely distributed in eastern Asia, has been consumed as a traditional Chinese medicine and food ingredient for centuries [6]. Currently, most peppermint leaves are dried and used as raw material for their natural flavor and peppermint essential oil extraction. However, *Mentha haplocalyx* leaves have a very short shelf life, and when it appears on the market in large quantities, changes in color and in the volatile components due to shrinkage and reduced moisture content can lead to defects in the texture and a decrease in quality [7]. To overcome this problem, various drying techniques including sun drying, microwave drying, hot air drying, infrared drying, and hybrid drying are commonly used to reduce the moisture content and extend the shelf life of aromatic plants [8,9,10,11]. The above different drying methods have been intensively studied in order to improve the quality of dried peppermint leaves by evaluating their effects on polyphenol content, antioxidant capacity, and the yield of peppermint essential oils [12,13]. Among them, hot air drying is recognized as an optimal approach for drying the plant-based raw food materials, based on the advantages of feasible drying equipment with low cost and high drying efficiency [14]. Its advantages are also due to weather resistance and controllable temperature compared to sun drying or drying in the shade. Accordingly, the hot air drying has been widely applied to dry aromatic and medicinal plants in non-tropical countries [15]. The impact of hot air-drying technology on quality attributes of foods has been studied in recent years. For instance, the changes of lipid composition and oxidation in pistachios, the effect of volatile components of the enoki mushroom as well as the physicochemical properties and biological activity of mushroom polysaccharides induced by hot air drying have been studied [16,17,18].

The drying of food products needs to fulfill not only consumer expectations but also processing requirements such as ease of handling and low equipment cost. The conventional drying process for *Mentha haplocalyx* leaves at the harvesting site consists of drying it in the shade, which possibly results in insufficient drying and loss of volatile bioactive components. Awareness of food safety and quality by consumers and the availability of affordable drying equipment has encouraged us to develop a drying approach that helps to retain higher nutrients and to feature bioactive constitutes. Emerging evidence has demonstrated that a hot air thin layer model has been successfully applied to process a variety of plant-based food materials, particularly vegetables and fruits [18,19,20,21]. Among those studies, various thin-layer drying models have been applied to demonstrate the thin-layer drying process, such as the Lewis, Page, Logarithmic, Midill and other models [22]. In a thin layer model, raw plant materials are spreading out in a thin layer with a mass of hot air flow transferring between each layer. As such, the efficiency of the drying process can be significantly improved to ensure a higher quality of dried peppermint leaves [23]. Nonetheless, there has been relatively little research on the drying characteristics of *Mentha haplocalyx* using hot air thin layer drying. Besides, knowledge the effects of the hot air thin layer drying process on the bioactive and sensory attributes of volatile extracts prepared by dried *Mentha haplocalyx* leaves is still scarce. These bioactive and sensory attributes are important key factors for the overall quality of the peppermint products. Thus, the present research aims to determine the drying kinetics of *Mentha haplocalyx* using hot air thin layer drying and to demonstrate the effect of the drying temperature on the microstructure of dried leaves and the quality of *Mentha haplocalyx* dried leaves, including their bioactive composition and sensory attributes.

## 2. Materials and Methods

### 2.1. Materials

The peppermint variety *Mentha haplocalyx* was purchased from Ji’an, Jiangxi Province, China, in August 2017. The collected samples were identified by Shouwen Zhang (Jiangxi University of Traditional Chinese Medicine, Nanchang, China) and stored in a cool place. The voucher specimens were stored at the Herbarium of Jiangxi University of Traditional Chinese Medicine.

### 2.2. Hot Air Thin Layer Drying of Mentha haplocalyx Leaves

Fresh *Mentha haplocalyx* leaves (100 g) were dried in a forced air oven at 35, 45 or 55 °C (BPG-9070A, 1500W, Yiheng Instruments Co., Ltd., Shanghai, China). The natural convection airflow was 2.0 m/s. The blast drying oven used in this work has a similar structure to that used by Hii and his colleagues [24]. The chopped fresh leaves were spread thinly in a single layer (2–3 cm thick) on a tray 15 cm in diameter. Heat was generated by a heater integrated in the oven wall. Exhaust gas escaped through the vents (diameter, 5 cm) at the rear of the oven. The tray was regularly removed and weighed by an electroin balance (FA2004A, Shanghai Precision Scientific Instruments Co., Shanghai, China) at intervals of 5 min (0–60 min), 10 min (60–180 min), 20 min (180–300 min), 30 min (300–420 min) and followed by 60 min until a constant weight was reached.

### 2.3. Microstructure Analysis of Dried Mentha haplocalyx Leaves

The influence of different pretreatments on the microstructure of dried samples was assessed by scanning electron microscopy (SEM; FEI-Quanta250, Waltham, MA, America). Samples were mounted on a double-sided tape and coated with a thin layer of gold film. The SEM imaging operation was performed at an acceleration voltage of 15 KV.

### 2.4. The Volatile Extracts Preparation 

The sample of original *Mentha haplocalyx* leaves were dried by air flow and evaluated by precision vane anemometer (AS-H, Avos High Precision Anemometer, Avos Co., Ltd., Shenzhen, China) after washing and stored in a cool ventilated place at 10 °C with a humidity of 40%. Subsequently, the above fresh samples and a series of samples dried by hot air at the different temperatures were prepared in the following manner. Briefly, the samples prepared from the fresh and dried peppermint leaves were chopped at different temperatures, with 100 g samples accurately weighed, placed in a sealed 2 L distillation flask (GC-17, 2L, SHUNIU), and then placed in a sealed container with distilled water (600 mL) and a small amount of zeolite. The above samples were immersed for 1 h and then heated at 100 °C for 3 h. The resulting distillate was dehydrated with anhydrous sodium sulfate for 30 min to obtain yellow volatile extracts, which were then transferred to a dry test tube and weighed. The collected volatile extracts were mainly essential oils, and were stored at 4 °C for further analysis. The essential oil yields were then analyzed in triplicate and the results were expressed as average values. The percentage volatile oil yield was calculated by dividing the weight of volatile oil by the weight of *Mentha haplocalyx* leaves. 

### 2.5. Determining Behavior of Hot Air Thin Layer Drying Process 

#### 2.5.1. Drying Rate

The drying rate (*DR*) of *Mentha haplocalyx* water loss refers to the ratio of the moisture content (dry basis) of the materials at two adjacent moments and the time interval, as calculated using Equation (1).
(1)$$DR=\frac{-(M_{t+\Delta t}-M_{t})}{\Delta t}$$
where *M*_(*t*+∆*t*)_ is the dry basis moisture content of the leaves at time *t* + ∆*t* (min) during the drying process.

#### 2.5.2. Mathematical Modeling of Thin Layer Drying Dynamics

To investigate the drying characteristics of the *Mentha haplocalyx* leaves, the drying behavior must be expressed as accurately as possible. In this study, the experimental drying data at different temperatures were fitted using seven classical layer drying models [25,26,27,28,29,30,31]. In these models, *MR* represents the dimensionless moisture ratio (dry basis), calculated using Equation (2):(2)$$MR=\frac{M_{t}-M_{e}}{M_{0}-M_{e}}$$
where *M_t_* is the moisture content of the stem, *M*_0_ is the initial moisture content of the fresh stem, and *M_e_* is the equilibrium moisture content. Since *M_e_* is small relative to *M*_0_ and *M_t_* and can be ignored, Formula (2) can be simplified to Formula (3).
(3)$$MR=\frac{M_{t}}{M_{0}}$$

Goodness was evaluated using statistical parameters such as the correlation coefficient (*R*^2^), chi-square (*χ*^2^), and root mean square error (*RMSE*). Generally, higher *R*^2^ and lower *χ*^2^ and *RMSE* values result in a better model fit. *χ*^2^ and *RMSE* were calculated using Equations (4) and (5), respectively:(4)$$\chi^{2}=\frac{\sum_{i=1}^{N}{\left(MR_{\exp,i}-MR_{pre,i}\right)}^{2}}{N-Z}$$
(5)$$RMSE=\sqrt{\frac{1}{N}\sum_{i=1}^{N}{\left(MR_{\exp,i}-MR_{pre,i}\right)}^{2}}$$
where *MR*_exp*,i*_ and *MR_pred,i_* are the *i*th experimental and predicted moisture ratios, respectively, *N* is the number of observations, and *z* is the number of parameters. In this work, nonlinear or linear regression analysis and statistical parameter calculations were performed using Originlab 8.0 software (OriginLab Corporation, Northampton, MA, USA) (Table 1).

#### 2.5.3. Effective Diffusivity and Activation Energy

The drying characteristics of biological products during the rate of decline can be described by Fick’s diffusion equation. The solution to this equation, developed by Crank, can be used for a variety of regular-shaped objects, such as rectangular, cylindrical, and spherical products. Equation (6) can be applied to particles with plate geometry by assuming a uniform initial moisture distribution [6,32]; where *D_eff_* is the effective diffusion coefficient (m^2^/s) and L0 is the half thickness (m) of the slab.
(6)$$MR=8\pi^{-2}\sum_{i=1}^{N}{\left(2n+1\right)}^{-2}\exp[-{\left(2n+1\right)}^{2}\pi^{2}D_{eff}tL^{-2}]$$

For drying over a long period, Equation (6) can be further simplified to retain only the first term and rewritten as Equation (7). In logarithmic form [32]. The diffusion coefficient can usually be determined by plotting experimental dry data as ln *MR* vs. drying time according to Equation (7) and calculating from the slope of the line.
(7)$$\ln MR=\ln(8/\pi^{2})-\pi^{2}D_{eff}t/L^{2}$$

The effective water diffusion coefficient can be temperature dependent according to the simple Arrhenius equation in Equation (8) [33]:(8)$$D_{\mathrm{eff}}=D_{0}\exp(-E_{a}/RT)$$
where *D_eff_* is the effective water diffusion coefficient (m^2^/s), *D*_0_ is the constant corresponding to the diffusion coefficient (m^2^/s) at infinite high temperature, *E_a_* is the activation energy (KJ/moL), and R is the general gas constant 8.314 KJ/moL, and *T* is the absolute temperature (K). The activation energy (*E_a_*) and constant D_0_ were determined by plotting ln (*D_eff_*) vs. 1/*T* after linearization using Equation (8).

### 2.6. Determination of Moisture Content

The moisture content was measured according to the standard air oven method [34]. The moisture content was calculated is using Equation (9):(9)$$M=\frac{W_{w}-W_{d}}{W_{d}}$$
where *M* is the water content of the *Mentha haplocalyx* leaf (dry basis), *W_w_* is the wet weight (g), and *W_d_* is the dry weight (g) of the sample. Experiments were performed in triplicate.

### 2.7. GC-MS Analysis of Volatile Extracts Derived from Dried Mentha haplocalyx Leaves

GC-MS analysis was performed on a 7890A gas chromatograph (Agilent, Palo Alto, CA, USA) equipped with a 5975C Plus mass spectrometer (Agilent, Palo Alto, California, American). A fused silica capillary Agilent Technology HP-5ms (5% phenyl methyl siloxane) column (30 m × 0.25 mm i.d.; film thickness, 0.25 μm) was used for the separation. The injector temperature was 250 °C and the detector temperature was 250 °C. The initial temperature was kept at 70 °C for 2 min, then gradually increased to 220 °C at a rate of 5 °C/min, and finally held at 220 °C for 5 min. The linear velocity of the helium carrier gas was 1.0 mL/min at a split ratio of 50:1. EI was used as the ion source and the ion source temperature was 230 °C. The sector mass analyzer was set to scan from 10 to 300 amu with a scan time of 1 s. Diluted samples (10 mg/mL) were prepared using dichloromethane and samples of 1.0 L were injected for analysis. Component identification was performed by matching the recorded mass spectra with standard mass spectra from the National Institute of Standards and Technology (NIST05.LIB) library data provided by the GC-MS system software, literature data, and standards of the main components. Quantitative analysis of each volatile oil component (expressed as area percentage) was conducted using peak area normalization measurements, and the results were given as mean values of three injections for each sample.

### 2.8. Electronic Nose Detection 

The flavor characteristics of the above samples prepared under different drying temperature were detected by using an electronic nose (I-Nose type electronic nose, Shanghai Refine Intelligent Technology Co., Ltd., Shanghai, China). A direct headspace aspiration method was applied, and the injection needle was inserted into the cup containing the sample powder. Each group of samples was measured three times in equilibrium. The sampling time was 120 s, the sensor rinsing time was 180 s, and the injection volume flow rate was 600 mL/min. Equilibrium measurement was performed three times. The signal sensor of the electronic nose consisted of 14 sensors, and their main characteristics are shown in Table 2. All data were analyzed statistically using OriginPro 2016 software (OriginLab Corporation, Northampton, MA, USA). radar analysis and principal component analysis (PCA) were performed using the software that came with the electronic nose.

### 2.9. Statistical Analysis

The statistical analysis was performed using Origin software. A one-way single factor Analysis of Variance (ANOVA) test was used for analyzing the drying kinetics of *Mentha haplocalyx* leaves and determining the effects of drying temperature on the bioactive and sensory attributes of their respective products.

## 3. Results and Discussion

### 3.1. Drying Kinetics of Mentha haplocalyx Leaves

In the present study, the hot air thin layer drying of *Mentha haplocalyx* leaves were carried out at 35 °C, 45 °C and 55 °C. The initial moisture content (dry basis) of the *Mentha haplocalyx* leaves was 5.10 ± 0.22 (mean ± SD). The relationship between the moisture content of the dry base of peppermint at a series of time points during the drying process and under the different hot air-drying temperatures are shown in Figure 1a. The optimal moisture content (w.b) level for storage of dried *Mentha haplocalyx* without risk of spoilage was less than 25% moisture [35]. In Figure 1a, the higher the temperature, the faster the rate of water loss and the shorter the time required for drying is shown. The initial dry base moisture content of peppermint decreased from 5.10 to a constant value of 0. Under this condition, the different drying times used in this study has been determined as 238 min at 55 °C, 424 min at 45 °C, and 1017 min at 35 °C, respectively, as shown in Figure 1a. The drying duration required at 35 °C was 4.27 times longer than that at 55 °C. It was noticed that the drying temperature is the key factor in determining the drying time. The higher drying temperature requires a greater heat power with a massive transportation of air that can shorter the drying duration [36]. Moreover, the relationship between the moisture ratio (*MR*) and drying time curves of peppermint at different drying temperatures in Figure 1b shows that the higher the temperature, the shorter time required for the drying process, while the *MR* values of peppermint leaves are the same during the drying process. As observed in Figure 1b, the changes of *MR* as well as the drying rate were found to agree with the finding of the previous study at the different drying temperatures [23]. These results suggested that the effect of air temperature is reflected with the drying rate for preparing dried *Mentha haplocalyx* leaves. 

Furthermore, as observed from the curves, the correlated slope indicates that the drying rate of peppermint leaves is fast in the early and middle stages, and tends to be gradually slowed down in the later stages (Figure 1). As the hot air temperature rose, the relative drying rate was increased, leading to accelerating moisture migration. In the early stages of drying, the slope of the drying curve initially increased, as shown in Figure 1c,d. This was caused by the internal and surface temperatures of the material increasing sharply and simultaneously. The surface moisture was evaporated and then diffused by heat, resulting in the surface temperature being lower than the internal temperature. Inconsistent external and internal temperatures creates an internal and external temperature gradience that forces the moisture migration to the outer surface. Hence, the driving force of internal moisture accelerates the diffusion of moisture to improve the drying efficiency [37]. Subsequently, the curves were gradually decelerated and ended in a constant drying rate under the different drying temperatures. The moisture decreases mainly occurred during the deceleration and drying stage. According to the literature, the slow-drying stage observed for *Mentha haplocalyx* leaves can be explained using the second law of thermodynamics, in which water moves from a higher water content to a lower water content and diffuses from the interior to the surface [38]. This is the main physical mechanism of moisture migration and water diffusion in the drying of various food and agricultural products, including *Mentha haplocalyx* leaves [23,39,40,41]. Our findings suggest that optimizing the drying temperature can help to shorten the drying time and to achieve a high active ingredient retention rate, maintain the color, and lower water activity [42]. Furthermore, there are progressive changes in physicochemical, textural and sensory properties with the changes of energy at the different drying stages that requires further investigation. 

In addition, the drying kinetics using various classical models were investigated in this study (Table 1 and Table 3). The statistical results are listed in Table 4. From the *R*^2^ values at different temperatures for the different models, *R*^2^ was above 0.99 in models 1, 2, 3, 4, 5 and 6 at all drying temperatures, except for model 7 at 45 °C and 55 °C. The results also showed that the models 1–6 fitted the experimental data well. Based on the average of the statistical parameters (MSP) of the six models at different temperatures, the highest MSP and the lowest *RMSE* in *R*^2^ appeared in model 1 (Midilli model). This finding suggested that the Midilli model perfectly represents the hot air thin layer drying process of *Mentha haplocalyx* leaves. Furthermore, recent findings have supported the finding that the Midilli model is effective in simulating the drying kinetics process of leafy plants [43,44]. Therefore, this model was selected in the present study to represent the drying characteristics of the *Mentha haplocalyx* leaves.

To predict the moisture ratio at any time under the specific drying condition, the parameters of the Midilli model at different drying temperatures were used to obtain an equation. To verify the represented model, the experimental moisture ratio was compared with the predicted values at various specific drying temperatures. The results showed that these experimental values were located near the curve, as shown in Figure 2a, indicating that the model effectively represented the drying progress of the *Mentha haplocalyx* leaves at drying temperatures of 35–55 °C. The effective diffusivity was determined after linearization of Equation (7) to give Equation (8). According to experimental ln *MR* regression between the falling rate period and corresponding time (s), when the temperatures were 35, 45 and 55 °C, the effective diffusion coefficients (*D_eff_*) were 7.51 × 10^−9^, 1.55 × 10^−8^, and 3.03 × 10^−8^ m^2^/s, respectively. The logarithm of *D_eff_* as a function of the reciprocal of the absolute temperature was plotted in Figure 2b. As is shown in Figure 2b, the slope of plotted *D_eff_* values was a straight line, indicating Arrhenius dependence. This result suggests that the plotted *D_eff_* values were perfectly fit in the linear regression. Our finding agreed with a previous observation in which the logarithm of *D_eff_* was also used to demonstrate a linear relationship with (1/*T*) [45]. According to the result shown in Figure 2b, the *R*^2^ value of regression was 0.999. In addition, the activity (*E_a_*) of moisture diffusion in drying *Mentha haplocalyx* leaves was estimated as 57.98 KJ/moL. The *E_a_* value seems slightly different compared to the previous finding in the thin layer drying of *Mentha spicata* leaves [23]. All in all, the above results confirmed that the Midilli model can be applied to describe the kinetics of hot air thin layer drying of *Mentha haplocalyx* leaves.

#### 3.1.1. The Effect of Hot Air Thin Layer Drying on Textural Property in *Mentha haplocalyx* Leaves

For consumers, the textural properties are a key feature of food products and a primary indication of food quality. The main reason is that the textural properties are correlated to the sensorial acceptance of products. Accordingly, the textures of dried *Mentha haplocalyx* leaves from different drying conditions were characterized by a scanning electron microscope (SEM). As shown in Figure 3, we observed the homogeneous textural characteristics with a similar appearance of dried samples among different drying conditions; but the texture of the dried samples is quite different compared to the fresh samples. In fresh sample, a full cell was well established with loosely distributed stomata (Figure 3). However, a significant shrinkage of epidermal cell shape was identified in the dried *Mentha haplocalyx* leaf cell, and condensed stomata were distributed on the surface (Figure 3). Among different drying conditions, the slight shrinkage was observed in the textural appearance of the cells from the thin layer dried sample under 35 °C, whereas the cell shrinkage of the samples dried at 45 and 55 °C are stronger than that in the dried sample at 35 °C, and the stomata are more densely distributed on the surface at 45 and 55 °C. The changes in the texture caused by the drying process have been observed in the previous study [46]. The alteration of the cellular membrane structure of the dried *Mentha haplocalyx* leaves might be caused by the loss of final moisture content during the drying process. 

#### 3.1.2. Effect of Drying Conditions on Volatile Bioactive Contents Derived from *Mentha haplocalyx* Leaves

The chromatographic separation conditions were obtained by systematically optimizing the temperature increase program. As shown in Figure 4, the separation of the bioactive components was measured and the peak shape was distributed well. The peaks were subjected to mass spectrometry to obtain the corresponding mass spectra. Data retrieved from the computer mass spectrometry data system (mass spectrometry database: NIST11 library), the retention times, and the related literature were combined to determine the composition of each component, with the results shown in Table 5 (corresponding to more than 85% of various compounds identified). Among these four samples, six identical components and 18 different components were found in their volatile extracts. The relative contents were calculated by normalization under the default integration setting. In this study, there were five different bioactive constitutes, including D-carvone, D-limonene, cineole, (Z)-13,7-dimethyl-3,6-octatriene, and caryophyllene, identified in the volatile extracts derived from the fresh *Mentha haplocalyx* leaves and their dried products at different temperatures. The significant changes in the content of bioactive compounds were identified in the volatile extracts derived from the fresh and dried samples at the different temperatures, which are the 4th, 6th, 16th and 20th components listed in Table 5. Firstly, a more than 2% change of content was detected in the 16th D-carvone with increasing drying temperatures of 35 °C, 45 °C and 55 °C, and the relative content in dried samples was 61.89%, 69.25% and 78.2%, respectively (Figure 5). D-carvone is a characteristic component rich in peppermint essential oil with a relative content of 70% [47]. A higher drying temperature may easily promote the release of volatile constitutes from the matrix of the leaf cell through water diffusion. Hence, the elevated release of D-carvone from the *Mentha haplocalyx* leaves was detected with the increasing drying temperature. However, a moderate decrease of D-Limonene was observed with increasing of drying temperature. Due to it containing olefin, D-limonene is easily oxidized during the hot air-drying process. The allylic oxidation of limonene at 6-position leads to the pyrolysis of glycol diacetate, after which it is saponified to carveol, which is further dehydrated into D-carvone [48,49]. As such, we observed that a transformation of D-limonene to D-carvone occurs in the present study. As shown in Figure 5, the content of D-Limonene in peppermint volatile extracts were 8.414%, 7.348% and 6.754% at drying temperatures of 35 °C, 45 °C and 55 °C, respectively. Eventually, the similar decreasing trend has been observed in the other two constitutes, including cineole (1.018%, 0.893% and 0.83%), and 1-caryophyllene (2.614%, 2.486% and 2.358%), with an increasing of drying temperature from 35 °C to 55 °C, respectively. The possible reason is that the above volatile components are heat sensitive. When the drying temperatures are high or the drying speed is too fast, damages or alterations of these compounds may occur.

Moreover, as expected with the aforementioned volatile bioactive components, alterations of other components have been detected at different drying conditions, as summarized in Table 5. To illustrate, the 23th 1,2,3,4,4a,7-Hexahydro-1,6-dimethyl-4-(1-methylethyl)-naphthalene and 18th carveol were only detected in fresh samples, but disappeared in the hot air dried samples. Besides, a series of volatile compounds such as β-Pinene, ocimene, α-gurjunene, [S-(E,E)]-1-methyl-5-methyl-8-(1-methylethyl)-1,6-cyclodecadiene became undetectable when increasing the heating temperature up to 55 °C. These results suggested that the hot air-drying process strongly affects the volatile constitutes of *Mentha haplocalyx* leaves, particularly when the level of these volatile constitutes are very low. Nonetheless, our finding indicated that most of the peppermint derived bioactive compounds had been well preserved in the hot air-dried samples, as listed in Table 5. Furthermore, transformation from compound 11 of (1S, 2S, 5S)-2-methyl-5-(1-methylethenyl) cyclohexanol to compound 10 of 2-methyl-5-(1-methylethenyl)-cyclohexanol was noticed after being heated over 35 °C. This indicated that structural changes of the volatile constitute may occur when the heating temperature is increased. A current study reported a decrease of aldehydes and alcohols with the loss of moisture content during the drying of pepper, and the level of esters was increased at the beginning and then continued to decline [50]. Findings from that study also suggested that high temperatures lead to enhancing the formation of ethyl octanoate, methyl octanoate, benzaldehyde, furan phenol, acetal, 5-methylfurfural and 2-acetylfuran, which agrees with our present observations. Another finding from drying *Thymus daenensis* leaves showed a decrease in the yield of essential oil with the increasing of the drying temperature [51]. All in all, the above results validated that a higher drying temperature might increase the release of volatile bioactive constitutes from the peppermint leaves, suggesting a potential impact of the high heating temperature on the sensory quality of the dried peppermint leaves. 

#### 3.1.3. Effect of Drying Conditions on Flavor Attributes of Dried *Mentha haplocalyx* Leaves

To determine the impact of the drying process on the sensory quality of the final product of *Mentha haplocalyx* leaves, the electronic nose has been applied in the present study to evaluate the volatile sensory profile in fresh and dried *Mentha haplocalyx* leaves. The electronic nose (E-nose) has been successfully used to study the sensory profile in a variety of dietary products, such as wine, tea, Perilla frutescens herbal extracts and throat lozenges [52,53,54,55]. The E-nose radar plots of the peppermint samples at different drying temperatures were shown in Figure 6a. The intensity of the sensor response results from the changes in the content of volatile compounds [56]. A total of 14 sensors’ characteristics of the E-nose were applied to evaluate the integrated sensory profile of the *Mentha haplocalyx* leaves as summarized in Table 2. As shown in Figure 6a, the intensity of sensor values in the E-nose radar plots indicated the enrichment and distribution of specific odor molecule distribution. Within the integrated sensory profile, S1 (aromatic compound sensitive) and S8 (amine sensitive) had significantly higher response values, followed by S2 (nitrogen oxide sensitive) and S5 (biosynthetic compound sensitive), while all other sensors had smaller response values for the peppermint samples. The present results demonstrated that the samples dried at 35 °C showed a relatively higher response in the S1, S2, S5 and S11 (volatile organic compounds) sections and was followed by drying at 45 °C and 55 °C, suggesting that the drying process significantly affected the above sensory features. The lowest aroma fingerprint data from the headspace of fresh *Mentha haplocalyx* leaves were observed in this study (Figure 6a), possibly due to their intact cell membrane that prevents the release of volatile compounds from the samples. Consequently, during the drying process, the volatility of different volatile components contained in *Mentha haplocalyx* was different. Moreover, the boiling point of the volatile compound and saturated vapor pressure play a determinant role in the volatility. As such, the volatile constitutes with a higher boiling point will be less easy to be volatilized during the hot air-drying process. The D-carvone of Mentha haplocalyx has a relatively higher boiling point (230 °C) compared to D-limonene (162.78 °C) [57,58]. The saturated vapor pressure of D-carvone (0.1 ± 0.5 mmHg) is lower than that of D-limonene (1.5 ± 0.2 mmHg) at room temperature, suggesting that the release of D-limonene is greater than that of D-carvone during hot air-drying process. Among different drying temperatures, an enhancing effect of the sensory response was observed in an air dry heating temperature at 35 °C compared to 45 °C and 55 °C. Since volatile organic compounds are more sensitive to heat, our results suggested that a moderate drying temperature would be beneficial to maintain the sensory properties of dried *Mentha haplocalyx* leaves.

A PCA analysis was performed to investigate the integrated impact of the heating temperature of hot air drying on the volatile compounds of dried *Mentha haplocalyx* leaves, as demonstrated in Figure 6b. In PCA, the contribution rates of PC1, PC2 and PC3 were 66.59%, 15.32% and 12.61%, respectively. The cumulative variance contribution rate was 94.52% (over 85%), indicating that the PCAwas able to reflect the sensory response information of the volatile compounds in *Mentha haplocalyx* leaves dried at different temperatures. As shown in the PCA result, the aromatic profiles analyzed by E-nose between the fresh and dried *Mentha haplocalyx* leaves at 35 °C were quite distinguished. That did not agree with our previous finding of the GC-MS analysis, which is possible because of the different way the samples were prepared. As shown in Figure 6b, even though a closed distribution was observed between hot air-dried samples at 45 °C and 55 °C, there was no clear overlap among the dried samples at the different temperatures, indicating that the profile of volatile aromatic compounds seems unique among the above *Mentha haplocalyx* leaf samples. This result suggested that the sensory attributes were distinguished by the volatile components that can be analyzed by the E-nose.

## 4. Conclusions

In the present study, hot air thin layer drying was applied to dry *Mentha haplocalyx* leaves at 35 °C, 45 °C and 55 °C. During the hot air-drying process, our results showed a higher temperature with a shorter time required for drying under the same *MR* value. In addition, various drying kinetic models have been assessed in this study to determine that the Midilli model is able to represent the hot air thin layer drying process. Due to the hot air-drying process playing a key role in the sensory quality of dried *Mentha haplocalyx* leaves, we subsequently explored the drying process-mediated textural and sensory alterations in *Mentha haplocalyx* leaves. As such, the SEM, GC–MS and E-nose technology were employed to study the effects of different drying temperatures on the textural and sensory characteristics of dried *Mentha haplocalyx* leaves. In the present study, an increasing degree of the cellular membrane shrinkage was detected during the drying process with the increasing temperatures. Furthermore, a total of 24 volatile constitutes were identified by GC-MS in fresh and dried samples, and the contents of some of them were dramatically decreased as the drying temperature was increased. An increasing content of D-carvone, which is the feature volatile component of peppermint, was identified to increase in this study from 61.89%, to 69.25% and then 78.2% when hot air drying at 35 °C, 45 °C and 55 °C, respectively. It revealed that other volatile compounds, including D-Limonene, cineole and l-caryophyllene, were reduced by the hot air-drying process. This finding indicated that a moderate drying temperature may contribute to enhancing the sensory quality of dried *Mentha haplocalyx* leaves. Eventually, both the flavor radar fingerprint and PCA of the E-nose was applied to evaluate the flavor profile in fresh and dried *Mentha haplocalyx* leaf samples at different drying temperatures. Findings from our results indicated that a relatively lower heating temperature of 35 °C has a less negative impact on the aromatic profile and related flavor quality. The underlying reason is that the volatile organic compounds are more sensitive to the heat during the drying process. The findings of this study provide a theoretical basis for the development of a hot air thin layer drying process for *Mentha haplocalyx* leaves and the improvement in its sensory qualities and related products. The future persepective is to continually improve the onsite drying technique for *Mentha haplocalyx* leaves and to develop a convenient monitor system to control the sensory quality of the final products based on the findings from our current study.

## Figures and Tables

**Figure 1 foods-11-00784-f001:**
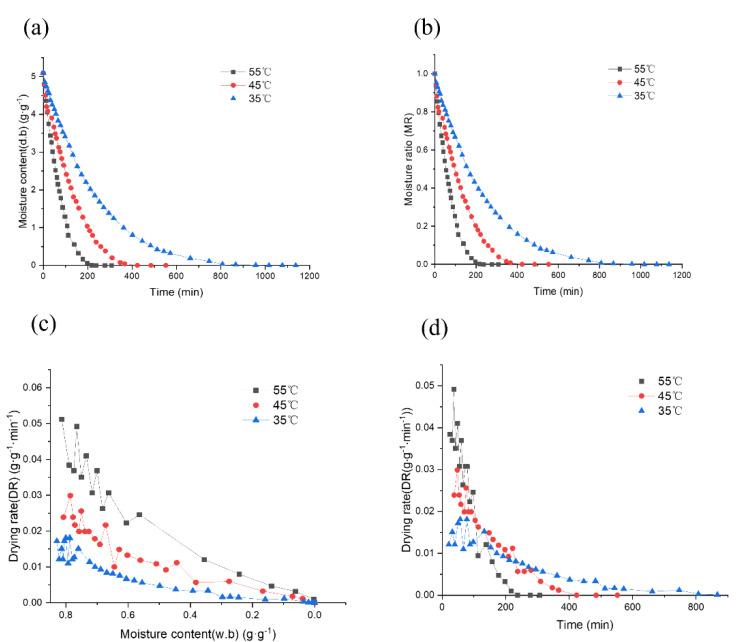
Drying kinetic curve of *Mentha haplocalyx*. (**a**) dry basis moisture content of *Mentha haplocalyx*, (**b**) the moisture ratio curve of *Mentha haplocalyx*, (**c**) drying rate with the wet basis of moisture content of the curve, (**d**) drying rate curves of *Mentha haplocalyx* at different drying temperatures.

**Figure 2 foods-11-00784-f002:**
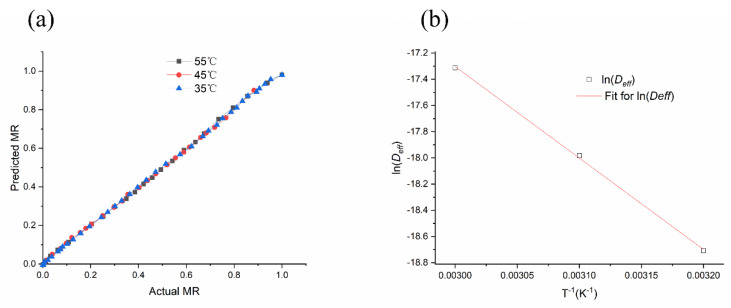
(**a**) The predicted *MR* by the Midilli model vs Actual *MR*, (**b**) Arrhenius type relationship between effective diffusivity and temperature.

**Figure 3 foods-11-00784-f003:**
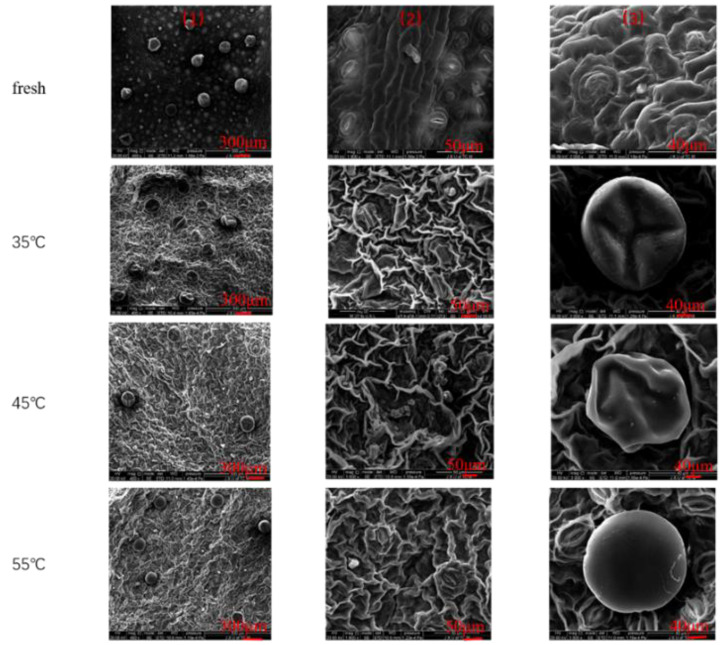
Microstructure diagrams of *Mentha haplocalyx* with different drying methods: (**1**) ×400 times magnification; (**2**) ×1600 times magnification; (**3**) ×3000 times magnification; fresh = fresh sample; 35 °C = 35 °C hot air drying; 45 °C = 45 °C hot air drying; 55 °C = 55 °C hot air drying.

**Figure 4 foods-11-00784-f004:**
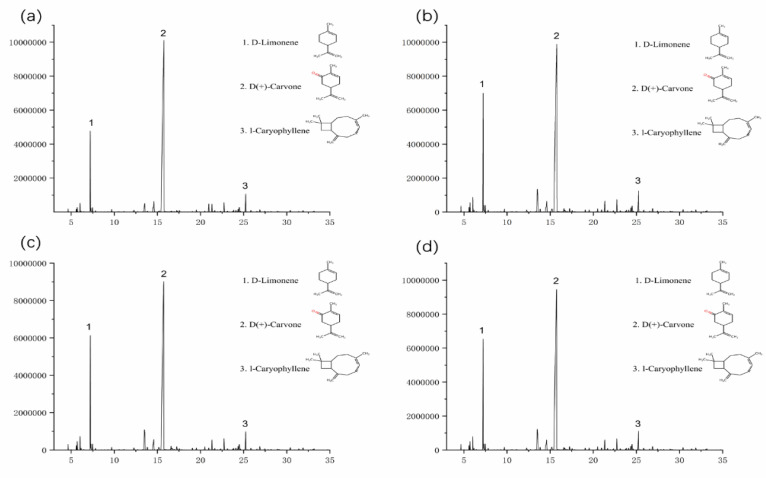
Total ion chromatogram of volatile oil component GC-MS before and after drying of *Mentha haplocalyx*. (**a**) Fresh *Mentha haplocalyx* volatile oil, (**b**) *Mentha haplocalyx* volatile oil after drying at 35 °C, (**c**) *Mentha haplocalyx* volatile oil after drying at 45 °C, (**d**) *Mentha haplocalyx* volatile oil after drying at 55 °C.

**Figure 5 foods-11-00784-f005:**
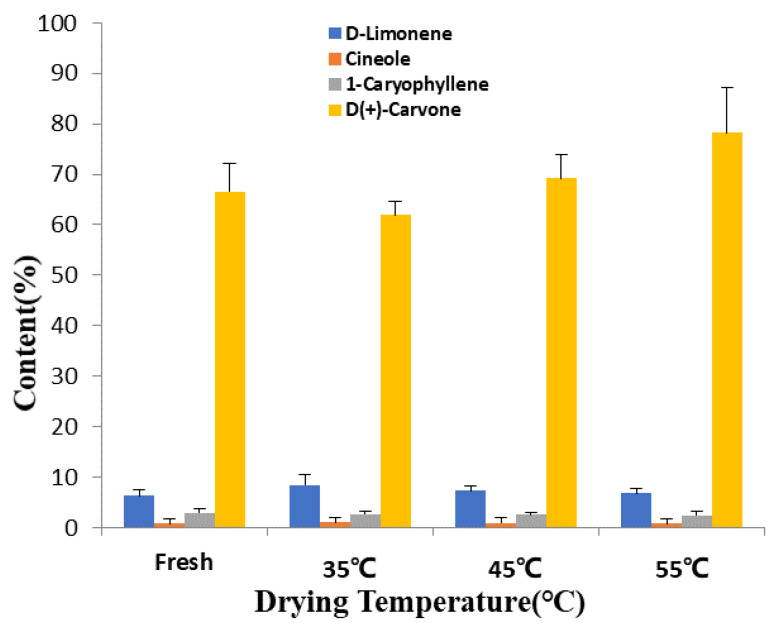
Contents of four common components in *Mentha haplocalyx* volatile oil at different temperatures.

**Figure 6 foods-11-00784-f006:**
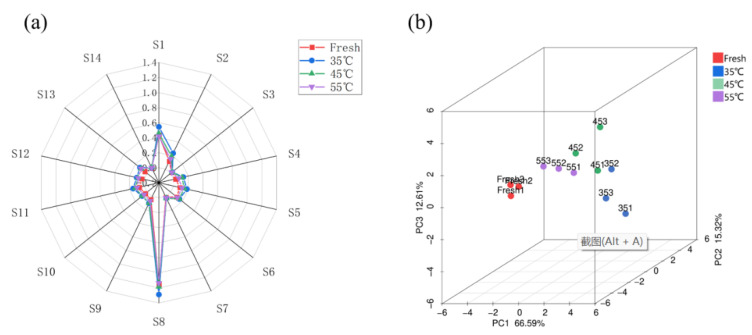
Radar fingerprint (**a**) and PCA (**b**) of the electronic nose of *Mentha haplocalyx* volatile components under different drying temperature conditions.

**Table 1 foods-11-00784-t001:** The typical thin-layer models selected for the *Mentha haplocalyx* drying curve.

No	Model Name	Model Equation	References
1	Midilli	*MR* = a exp(−kt^y^)+bt	[25]
2	Page	*MR* = exp(−kt^y^)	[26]
3	Overhults	*MR* = exp[−(kt^y^)]	[27]
4	Modified Page	*MR* = a exp[−(kt)^y^]	[28]
5	Logaritmic	*MR* = a exp(−kt)+c	[29]
6	Two terms Exponential	*MR* = a exp(−kt)+(1 − a) exp(−kat)	[30]
7	Newton	*MR* = exp(−kt)	[31]

Note: t. Drying time (s); a, k, b, y, c. Model coefficients.

**Table 2 foods-11-00784-t002:** Main sensors’ characteristics of I-Nose electronic nose instruments.

	I-Nose Eletronic Nose
Sensors	Sensors characteristics
Sn1	More sensitive to aromatic compounds
Sn2	More sensitive to nitrogen oxides
Sn3	More sensitive to sulfides
Sn4	More sensitive to organic acids, terpenoids
Sn5	More sensitive to biosynthetic compounds
Sn6	More sensitive to thionine
Sn7	More sensitive to aliphatic hydrocarbons
Sn8	More sensitive to amines
Sn9	More sensitive to dihydrostyrene
Sn10	More sensitive to hydrocarbons
Sn11	More sensitive to volatile organic compounds
Sn12	More sensitive to sulfides
Sn13	More sensitive to ethylene
Sn14	More sensitive to volatile gases generated during cooking

**Table 3 foods-11-00784-t003:** Model coefficients of the Midilli model at different drying temperature.

Temperature	Model Coefficients			
	a	k	y	b
35 °C	0.97999	0.0028	1.07691	−0.00001
45 °C	0.94597	0.00347	1.13413	−0.00014
55 °C	0.98146	0.00688	1.14454	−0.00019

Note: a, k, b, y. Model coefficients.

**Table 4 foods-11-00784-t004:** Statistical parameters for various models.

Model	SP	35 °C	45 °C	55 °C	MSP
1	*R* ^2^	0.9997	0.9958	0.9983	0.9979
	*χ* ^2^	0.0000	0.0004	0.0002	0.0002
	*RMSE*	0.0062	0.0199	0.0128	0.0130
2	*R* ^2^	0.9991	0.9900	0.9963	0.9952
	*χ* ^2^	0.0001	0.0009	0.0004	0.0005
	*RMSE*	0.0099	0.0308	0.0190	0.0199
3	*R* ^2^	0.9991	0.9900	0.9963	0.9952
	*χ* ^2^	0.0001	0.0009	0.0004	0.0005
	*RMSE*	0.0099	0.0308	0.0190	0.0199
4	*R* ^2^	0.9996	0.9945	0.9970	0.9970
	*χ* ^2^	0.0001	0.0005	0.0003	0.0003
	*RMSE*	0.0072	0.0229	0.0171	0.0157
5	*R* ^2^	0.9994	0.9957	0.9977	0.9976
	*χ* ^2^	0.0001	0.0004	0.0002	0.0002
	*RMSE*	0.0082	0.0202	0.0150	0.0145
6	*R* ^2^	0.9994	0.9913	0.9965	0.9957
	*χ* ^2^	0.0001	0.0008	0.0003	0.0004
	*RMSE*	0.0085	0.0288	0.0185	0.0186
7	*R* ^2^	0.9985	0.9880	0.9893	0.9919
	*χ* ^2^	0.0002	0.0011	0.0011	0.0008
	*RMSE*	0.0130	0.0338	0.0324	0.0264

**Table 5 foods-11-00784-t005:** The active ingredient content of the Mentha haplocalyx at various temperatures by GC-MS.

No.	t_R_(min)	Formula	Compound	Common Name	Fresh	35 °C	45 °C	55 °C
1	5.249	C_10_H_16_	β-Pinene	\	0.46	0.39	0.455	\
2	5.479	C_10_H_16_	Myrcene	\	0.443	\	0.568	\
3	5.568	C_8_H_18_O	3-Octanol	\	\	0.228	\	\
4	7.362	C_10_H_16_	Cyclohexene,1-methyl-4-(1-methylethenyl)-, (4R)-	D-Limonene	6.306	8.414	7.348	6.754
5	7.431	C_10_H_18_O	1,3,3-Trimethyl-2-oxabicyclo[2 .2.2]octane	Cineole	0.8	1.018	0.893	0.83
6	7.531	C_10_H_16_	(Z)-13,7-dimethyl-3,6-octatriene	\	0.766	0.998	0.898	0.794
7	7.786	C_10_H_16_	Ocimene	\	0.3	0.4	0.35	\
8	9.085	C_10_H_18_O	2,6-Dimethylocta-2,7-dien-6-ol	Linalool	\	0.25	\	\
9	9.872	C_10_H_18_O	Endo-1,7,7-trimethyl-bicyclo[2.2.1]heptan-2-ol	Borneol	\	0.29	\	\
10	10.617	C_10_H_18_O	2-methyl-5-(1-methylethenyl)-Cyclohexanol	1,6-Dihydrocarveol	\	\	2.665	2.603
11	12.618	C_10_H_18_O	Cyclohexanol,2-methyl-5-(1-methylethenyl)-, (1S,2S,5S)-	\	4.35	4.395	\	\
12	13.671	C_9_H_16_O	Cyclopentanone,2-methyl-5-(1-methylethyl)-	\	3.258	5.428	5.634	4.724
13	13.878	C_10_H_16_O	(5R)-5-Isopropenyl-2-methyl-2-cyclohexen-1-ol	\	0.335	0.516	0.54	\
14	14.267	C_10_H_14_O	2,4-Cycloheptadien-1-one,2,6,6-trimethyl-	\	3.5	0.632	0.474	\
15	14.61	C_10_H_16_O	2-methyl-5-(1-methylethenyl)-2-Cyclohexen-1-ol	Dihydrocarvone	\	0.454	0.4	\
16	15.976	C_10_H_14_O	D-1-Methyl-4-isopropenyl-6-cyclohexen-2-one	D(+)-Carvone	66.55	61.89	69.25	78.2
17	18.127	C_10_H_12_O	cis-Anethol	\	1.015	1.385	\	1.12
18	19.023	C_10_H_16_O	2-methyl-5-(1-methylethenyl)-2-Cyclohexen-1-ol	Carveol	0.525	\	\	\
19	22.263	C_15_H_24_	α-Gurjunene	\	0.315	0.32	0.36	\
20	25.516	C_15_H_24_	l-Caryophyllene	\	2.874	2.614	2.486	2.358
21	26.368	C_15_H_24_	1,4,8-Cycloundecatriene,2,6,6,9-tetramethyl-, (1E,4E,8E)-	α-Caryophyllene	0.89	0.827	0.747	\
22	28.421	C_15_H_24_	[S-(E,E)]-1-methyl-5-methyl-8-(1-methylethyl) 1,6-cyclodecadiene		1.78	2.032	1.882	\
23	30.208	C_15_H_24_	Naphthalene,1,2,3,4,4α,7-hexahydro-1,6-dimethyl-4-(1-methylethyl)-		0.398	\	\	\
24	47.105	C_15_H_26_O	1-Naphthalenol,1,2,3,4,4α,7,8,8α-octahydro-1,6-dimethyl-4-(1-methylethyl)-, (1R,4S,4αR,8αR)-	l-α-Cadinol	\	0.849	1.14	1.22

## Data Availability

Data is contained within the article.

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
