# Peer review of "Assessment of Drying Kinetics, Textural and Aroma Attributes of *Mentha haplocalyx* Leaves during the Hot Air Thin-Layer Drying Process"

_foods, 2022, doi:10.3390/foods11060784_

Round 1

Reviewer 1 Report

The revised article addresses a topic that has been widely studied from the physical point of view, however the study of bioactive compounds that can be affected during the drying process is interesting. The studies for the identification and quantification of the compounds were carried out in a good way.

I consider that in the discussion of the results it is necessary to reinforce with showing and comparing what has been found in other studies that explain why certain phenomena occur, especially in the case of the increase of certain bioactive compounds.

Within the attached document you will find more observations made to the document. Otherwise, the article meets the appropriate quality for a high-impact journal.

Author Response

Response: Thank you for the valuable comment. We agree to add more information to discuss the phenomena observed in the present study. Our modifications have been added into the manuscript (Line 408-412) and (Line 484-491). In this study, we observed a phenomenon is that a carvone was increased with an increasing temperature, while D-limonene and carveol were decreased. The following demonstration elucidates the possible underlying mechanism. The internal olefin is a feature structure of D-limonene which is easily oxidized during the hot air-drying process. The allylic oxidation of limonene at 6-position leads to pyrolysis of glycol diacetate and then saponified to carveol, which is further dehydrated into carvone [1, 2]. As such, we observed that a transformation of D-limonene to carvone occurs in the present study. Moreover, during the drying process, the volatility of different volatile components contained in Mentha haplocalyx was different. The boiling point of the volatile compounds and saturated vapor pressure play a determinant role in volatility. As such, the volatile constitutes with a higher boiling point will be less easy to be volatilized during the hot air-drying process. The carvone of Mentha haplocalyx has a relatively higher boiling point (230oC) compared to D-limonene (162.78 oC) [3, 4]. In addition, the saturated vapor pressure of carvone (0.1±0.5 mmHg) is lower than that of D-limonene (1.5±0.2 mmHg) at room temperature, suggesting that the loss of D-limonene during the hot air-drying process is more than carvone. Taking together, our finding indicated that as elevation of drying temperature, the carvone content in Mentha haplocalyx was increased through the oxidative transportation from D-limonene. 

  1. Wróblewska, A., The Epoxidation of Limonene over the TS-1 and Ti-SBA-15 Catalysts. Molecules, 2014. 19(12): p. 19907-19922.
  2. Linder, S.M. and F.P. Greenspan, Reactions of Limonene Monoxide. The Synthesis of Carvone. The Journal of Organic Chemistry, 1957. 22(8): p. 949-951.
  3. Liu, S.X. and P.K. Mamidipally, Quality Comparison of Rice Bran Oil Extracted with d-Limonene and Hexane. Cereal Chemistry, 2005. 82(2): p. 209-215.
  4. Schlyter, F., et al., Carvone and less volatile analogues as repellent and deterrent antifeedants against the pine weevil, Hylobius abietis. Journal of Applied Entomology, 2004. 128(9-10): p. 610-619.

Reviewer 2 Report

The authors presented a well-written manuscript in which novel results were revealed. However, minor modifications are needed. Please refer to the following comments.

Abstract section:

Please state the best model that described the drying kinetics of the product. 

Introduction section:

For matimathical models, more refrences are needed.

Material and method section:

Line 107: Please state how the authors measure the air flow and include the model and the country of the device.

Line 109: How much the power of the heater used?

Line 111: Please insert the details of the  balance used for weighning the samples.  What are the time intervals during which the weight was measured?

Line 112: The subtitle “Textural characterization of dried Mentha haplocalyx leaves” should be replaced with “ Microstructure Analysis”.

Line 118: What type of the device that measured the air flow?

Line 119: What were the conditions of the cool ventilated place eg, T and RH?

Line 123: Please include the details of the container used.

Line 137: Please include the unit of the time.

Line 185: Please insert the units of the weight.

Results section:

Line 244: How did you determine that the optimal moisture content (w.b) level for storage of Mentha haplocalyx without risk of spoilage was less than 0.13?

Table 2 should be moved to the method section.

Figure 5: Standard deviations and Error bars are needed.

Conclusion section:

Future studies should be included

Author Response

The authors presented a well-written manuscript in which novel results were revealed. However, minor modifications are needed. Please refer to the following comments.

We appreciate for reviewers’ comment and all our modification in the manuscript has been highlighted in light blue.

Abstract section:

Please state the best model that described the drying kinetics of the product. 

Response: Thank you for the valuable comment. The best model determined in this study has been indicated in the abstract section.

Introduction section:

For matimathical models, more refrences are needed.

Response: Thank you for the valuable comment. According to this comment, one review has been cited in this study to demonstrate different matimathical models (Line 96-98).

Material and method section:

Line 107: Please state how the authors measure the air flow and include the model and the country of the device.

Response: Thank you for the comment. According to this comment, the instrument information was added (Line 136-137).

Line 109: How much the power of the heater used?

Response: Thank you for the valuable comment.The information has been added in Line 120.

Line 111: Please insert the details of the balance used for weighning the samples.  What are the time intervals during which the weight was measured?

Response: Thank you for the comment. According to this comment, the way of weighting measurement has been illustrated in Line125-128.

Line 112: The subtitle “Textural characterization of dried Mentha haplocalyx leaves” should be replaced with “ Microstructure Analysis”.

Response: Thank you for the comment. According to this comment, the change has been made.

Line 118: What type of the device that measured the air flow?

Response: Thank you for the comment.

Line 119: What were the conditions of the cool ventilated place eg, T and RH?

Response: Thank you for the comment. According to this comment, the change has been added in Line 135-136.

Line 123: Please include the details of the container used.

Response: Thank you for the comment. According to this comment, the container information has been added.

Line 137: Please include the unit of the time.

Response: Thank you for the comment. The time unit (min) has been inserted in Line156.

Line 185: Please insert the units of the weight.

Response: Thank you for the comment. The weight unit (g) has been inserted in Line208.

Results section:

Line 244: How did you determine that the optimal moisture content (w.b) level for storage of Mentha haplocalyx without risk of spoilage was less than 0.13?

Response: Thank you for the valuable comment. According to the suggestion, we reviewed the current publication and found out that a water activity level less than 0.6 can effectively prevent the risk of spoilage for the storage of foods [5]. Thus, the moisture content of dried foods, including dried Mentha haplocalyx leaves, should be less than 25%, and aw is lower than 0.6 (Line 268-270).

Table 2 should be moved to the method section.

Response: Thank you for the comment. Table 2 has been moved to the method section (Line 179).

Figure 5: Standard deviations and Error bars are needed.

Response: Thank you for the comment. Figure 5 has been modified.

Conclusion section:Future studies should be included

Response: Thank you for the comment. Future studies have been added to the manuscript (Line 544-546).

  1. Food Preservation by Reducing Water Activity, in Food Microbiology: Principles into Practice. 2016. p. 44-58.
